# Integrating between-host transmission and within-host immunity to analyze the impact of varicella vaccination on zoster

Benson Ogunjimi[1,2]*, Lander Willem[1,2,3], Philippe Beutels[1,2,4], Niel Hens[1,2]

[1]Centre for Health Economics Research and Modeling Infectious Diseases, Vaccine and Infectious Disease Institute, University of Antwerp, Antwerp, Belgium; [2]Interuniversity Institute for Biostatistics and Statistical Bioinformatics, Hasselt University, Hasselt, Belgium; [3]Department of Mathematics and Computer Science, University of Antwerp, Antwerp, Belgium; [4]School of Public Health and Community Medicine, University of New South Wales, Sydney, Australia

**Abstract** Varicella-zoster virus (VZV) causes chickenpox and reactivation of latent VZV causes herpes zoster (HZ). VZV reactivation is subject to the opposing mechanisms of declining and boosted VZV-specific cellular mediated immunity (CMI). A reduction in exogenous re-exposure 'opportunities' through universal chickenpox vaccination could therefore lead to an increase in HZ incidence. We present the first individual-based model that integrates within-host data on VZV-CMI and between-host transmission data to simulate HZ incidence. This model allows estimating currently unknown pivotal biomedical parameters, including the duration of exogenous boosting at 2 years, with a peak threefold to fourfold increase of VZV-CMI; the VZV weekly reactivation probability at 5% and VZV subclinical reactivation having no effect on VZV-CMI. A 100% effective chickenpox vaccine given to 1 year olds would cause a 1.75 times peak increase in HZ 31 years after implementation. This increase is predicted to occur mainly in younger age groups than is currently assumed.

*For correspondence: benson.ogunjimi@uantwerp.be

## Introduction

Varicella-zoster virus (VZV) causes the itching, erythematous vesicular disease called varicella or chickenpox, mainly during childhood, and remains latent in neural ganglia afterwards. Latency can then be interrupted by episodes of (primarily subclinical) reactivation of VZV as shown by the detection of VZV in saliva or blood from otherwise healthy individuals (*Schünemann et al., 1998*; *Nagel et al., 2011*). VZV reactivations may also cause herpes zoster (HZ), which presents clinically as a painful dermatomal rash. HZ occurs most frequently in individuals with a drastically declined cellular immune status (*Dolin et al., 1978*), but ageing itself is often assumed to substantially reduce resilience of the VZV-specific immune response (*Miller, 1980*; *Berger et al., 1981*; *Levin et al., 2003*, *2008*). Recent research supports the hypothesis that waning of VZV cellular mediated immunity (CMI) by age is also influenced by cytomegalovirus (CMV) infection (*Ogunjimi et al., 2014*).

Effective and safe pediatric vaccines against varicella exist and have been universally implemented in some countries including the US, Australia, Greece, Germany, Japan and Taiwan (*Marin et al., 2008*). However, in many other countries policy makers have been hesitant to introduce childhood VZV vaccination due to the general population's perception of varicella as a relatively mild disease, as well as the so-called exogenous boosting hypothesis (*Hope-Simpson, 1965*;

**eLife digest** The itchy-scratchy misery of a chickenpox was until recently a rite of passage for children around the world. The varicella-zoster virus causes chickenpox infections. This virus persists in small numbers in nerve cells for many years after infection, and can reactivate from these cells. Often this reactivation causes no symptoms, but sometimes it results in a painful skin condition called shingles (or herpes zoster), especially in older adults.

Some countries—including the United States, Australia, Taiwan and Greece—have virtually wiped out childhood cases of chickenpox by requiring that children be vaccinated against the varicella-zoster virus. But some countries have hesitated. One reason for this hesitation is that exposure to individuals with a chickenpox infection helps boost the immunity of individuals who have previously been infected. This may help reduce the likelihood of these people developing shingles later in life. So, some countries have worried that chickenpox vaccinations might inadvertently increase the number of shingles cases. To assess this risk, many scientists have created computer models, but the models have some limitations.

Now, Ogunjimi et al. report a new individual-based model to assess the effect of childhood varicella vaccination on shingles cases that factors in the immune responses to varicella infection. The model suggests that re-exposure to the varicella virus through contact with infected people would only provide extra protection for about two years; this is much shorter than previous predictions that suggested it might last 20 years. The model also predicts that implementing a varicella vaccination program for children would almost double the number of shingles cases 31 years later. But this increase would be temporary.

The predicted increase in shingles cases is likely to disproportionately occur among 31- to 40-year-olds. This is unexpected because most previous models predict that older age groups would bear the brunt of a rise in shingles, but this younger population would be less likely to develop lasting complications of shingles. Together, these findings may allay some fears about implementing childhood varicella vaccination programs by showing that the benefits of re-exposure are limited.

Ogunjimi et al., 2013). This hypothesis is based on the concept of the secondary immune response and assumes that boosting of VZV-CMI occurs upon re-exposure to varicella. Boosted VZV-specific cellular immunity would subsequently reduce the risk of VZV reactivation and thence HZ. The introduction of widespread childhood VZV vaccination would reduce opportunities for varicella re-exposure and could therefore increase HZ incidence, as shown in model-based projections (*Schuette and Hethcote, 1999*; *Brisson et al., 2000*; *Bilcke et al., 2013*). Although current data on HZ-incidence post introduction of childhood VZV vaccination has caused controversy, a systematic review rating the quality of the evidence, concluded that exogenous boosting exists, but that its population-wide effect after widespread childhood VZV vaccination remains highly uncertain (*Ogunjimi et al., 2013*). One of the main points of criticism on current predictions by deterministic VZV simulation models is the entanglement of exogenous boosting, waning of immunity, immunosenescence and reactivation undermining the possibility of estimating the magnitude and duration of exogenous boosting accurately. A concern is that these entangled parameters can be chosen to fit these models well to observed HZ incidence data, but that they are too artificial to allow real and verifiable biological interpretations. This leads to currently unverifiable and potentially poor predictions of VZV dynamics beyond the fitted equilibrium states. An additional, hitherto ignored uncertainty, is the potential occurrence of endogenous boosting by subclinical VZV reactivation. Indeed, some studies have shown VZV to reactivate subclinically both in healthy individuals, immunocompromised and stressed individuals (*Schünemann et al., 1997*; *Mehta et al., 2003*; *Nagel et al., 2011*). However, the effect on VZV-CMI has not yet been quantified.

In the current paper, we describe the first individual-based VZV model, explicitly combining within and between-host dynamics. This model is based on experimental viro-immunological data and allows an accurate estimation of the exogenous boosting characteristics and explicit insertion or validation of experimental data.

# Results

## VZV IBM parameter prediction

Using a step-wise algorithm we initially found eight unique parameter sets leading to a reasonable fit of Belgian HZ incidence data (see *Table 1* and *Figure 1*). All best-fitting parameter sets were based on boosting scenario 3 (predefined exponential loss of boosted VZV-CMI). Seven of these models include exogenous boosting (defined as an exponential decay) with a peak value of 1.3, a boosting duration ranging between 2 and 15 years, no endogenous boosting, a weekly VZV reactivation probability and an annual VZV-CMI loss (= waning) estimated to be 1–1.5% and 1–2%, respectively. We note that although the fits are excellent for the most relevant age groups, 25 years and older, they are less suited to predict HZ incidence for younger ages. One model predicted a peak boosting value of 2.5, a boosting duration of only 1 year and no endogenous boosting. Although this model was less suited to fit to HZ data for older age groups, it fitted much better to HZ data for younger age groups. In additional analyses relaxing on some parameter constraints, we obtained final parameter sets that led to excellent predictions across all age groups (see *Table 1* and *Figure 2*). The two best fitting parameter sets predicted peak boosting values between 2.8 and 4 (maximum allowable value), a duration of boosting limited to 1 or 2 years and, as before, no endogenous boosting. Averaged VZV-specific CMI levels are shown in *Figure 3*.

## Childhood varicella vaccination and its implications for HZ incidence

Using our best fitting parameter sets (set 9 and 13 from *Table 1*) we simulated the effects of a 100% permanently effective varicella vaccine given to all children aged 1 year (100% coverage). Our predictions (averaged per parameter set) showed a net increase in HZ incidence during approximately 50 years (see *Figure 4*). The peak increase in HZ incidence was reached approximately 31 years after programme initiation and was 1.75 (1.64–1.87) (100% CI) times higher than the HZ incidence prior to varicella vaccination. The CI is constructed by using three runs per parameter set and normalizing each

**Table 1**. Best fitting parameter sets

| Parameter set | Deviance* | Annual waning rate (%) | Boosting scenario | Duration of boosting (years) | Peak fold increase following exogenous boosting | VZV weekly reactivation probability (%) | Distribution threshold VZV-CMI for HZ | Peak fold increase following endogenous boosting |
|---|---|---|---|---|---|---|---|---|
| Original Search (obtained after Step 2 in *Table 1*) | | | | | | | | |
| 1 | 926 | 2.0 | 3 | 10 | 1.3 | 1.5 | 4 | 1 |
| 2 | 939 | 1.5 | 3 | 3 | 1.3 | 1.5 | 4 | 1 |
| 3 | 949 | 2.0 | 3 | 7 | 1.3 | 1.5 | 4 | 1 |
| 4 | 951 | 2.0 | 3 | 12 | 1.3 | 1.5 | 4 | 1 |
| 5 | 968 | 2.0 | 3 | 7 | 1.3 | 1.0 | 4 | 1 |
| 6 | 970 | 1.0 | 3 | 2 | 1.3 | 1.0 | 4 | 1 |
| 7 | 969 | 2.0 | 3 | 15 | 1.3 | 1.5 | 4 | 1 |
| 8 | 934 | 1.0 | 3 | 1 | 2.5 | 5.0 | 4 | 1 |
| Border search | | | | | | | | |
| 9 | 751 | 1.0 | 3 | 1 | 2.8 | 5.0 | 4 | 1 |
| 10 | 799 | 1.0 | 3 | 1 | 3.1 | 5.0 | 4 | 1 |
| 11 | 965 | 1.5 | 3 | 2 | 3.4 | 5.0 | 4 | 1 |
| 12 | 804 | 1.5 | 3 | 2 | 3.7 | 5.0 | 4 | 1 |
| 13 | 722 | 1.5 | 3 | 2 | 4.0 | 5.0 | 4 | 1 |

*Results shown are averaged results per parameter set.

VZV, varicella-zoster virus; HZ, herpes zoster.

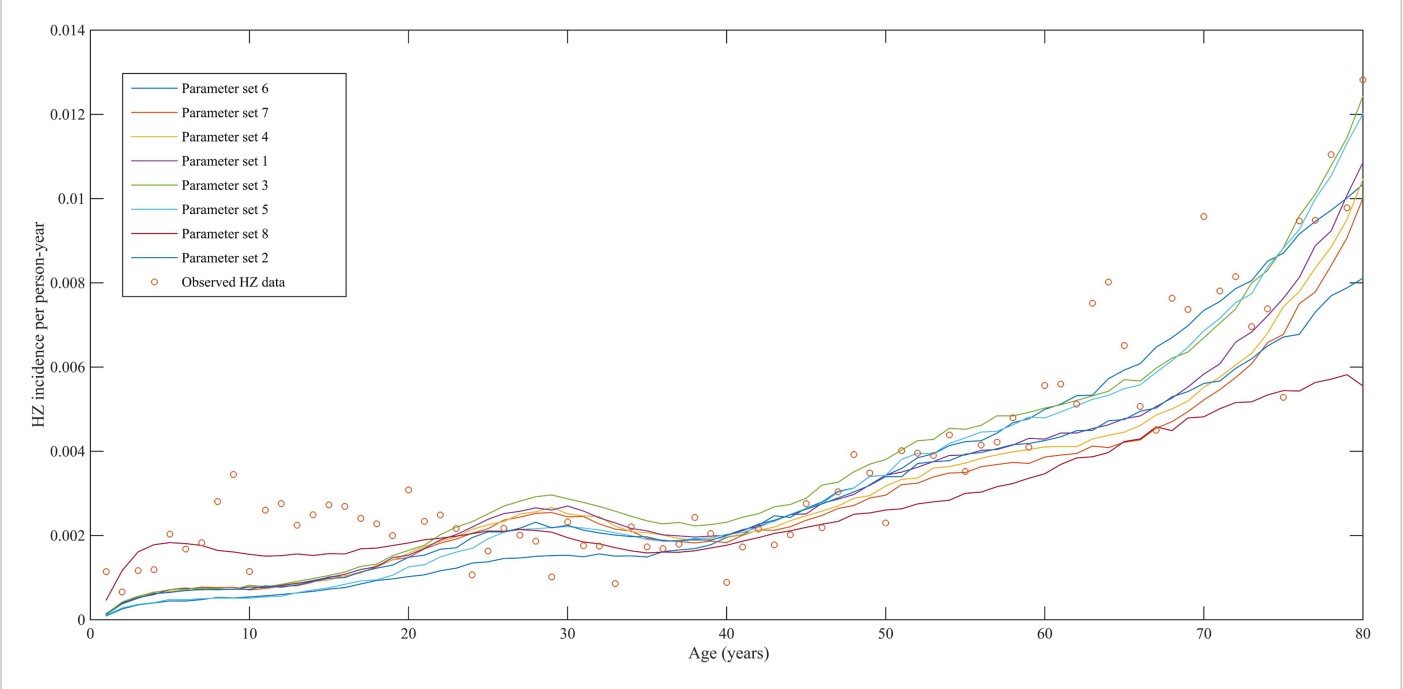

**Figure 1**. Observed (open circles) and simulated (continuous lines) Belgian herpes zoster (HZ) incidence data by age.

The following source data is available for figure 1:

**Source data 1**. Observed Belgian HZ incidence per age group and per person-year.

run by means of the run-specific averaged equilibrium HZ incidence between 100 and 300 years since the beginning of the simulation. This means that per 1,000,000 person-years the total number of HZ cases would increase on average from 3219 to 5313 (set 9) or from 3209 to 5955 (set 13). *Figure 5* shows that the relative contribution of different age groups to HZ incidence changes in the time period after introduction of varicella vaccination. Noteworthy is the relative dominance of the 31–40 year old age group between 10–50 years after introduction of varicella vaccination.

## Discussion

It is hypothesized that exogenous re-exposure to varicella increases VZV-specific cellular immunity (VZV-CMI). This natural consequence of the secondary immune response will reduce an individual's risk for HZ. When the probability of contact per unit-time between currently infectious and previously recovered varicella patients reduces, through a reduction in varicella incidence, it can be expected that HZ incidence temporarily increases due to a lack of exogenous boosting (*Hope-Simpson, 1965*; *Ogunjimi et al., 2013*).

To our knowledge, this study is the first to use an individual-based dynamic transmission model for VZV. This model allowed us to combine immunological and virological data to estimate key parameters in VZV population dynamics, such as the peak CMI response following re-exposure, duration of boosting and VZV subclinical reactivation characteristics. This means that in contrast to the deterministic models where abstract and ad hoc compartments are created to define the transition between different epidemiological states (for example the transition of a varicella recovered state to a zoster susceptible state), we can actually work with true biological, and verifiable, concepts such as VZV-CMI. Our best fitting parameter sets for Belgium suggest the effective duration of exogenous boosting to last only between 1 and 2 years. These predictions are significantly lower than those from the highly cited deterministic VZV model by Brisson et al., in which the average duration was predicted to be 20 years ([7–41 years] 95%CI) (*Brisson et al., 2002*). Our peak immunological response was estimated to be 2.8–4.0 times larger than the pre-re-exposure value. These predictions

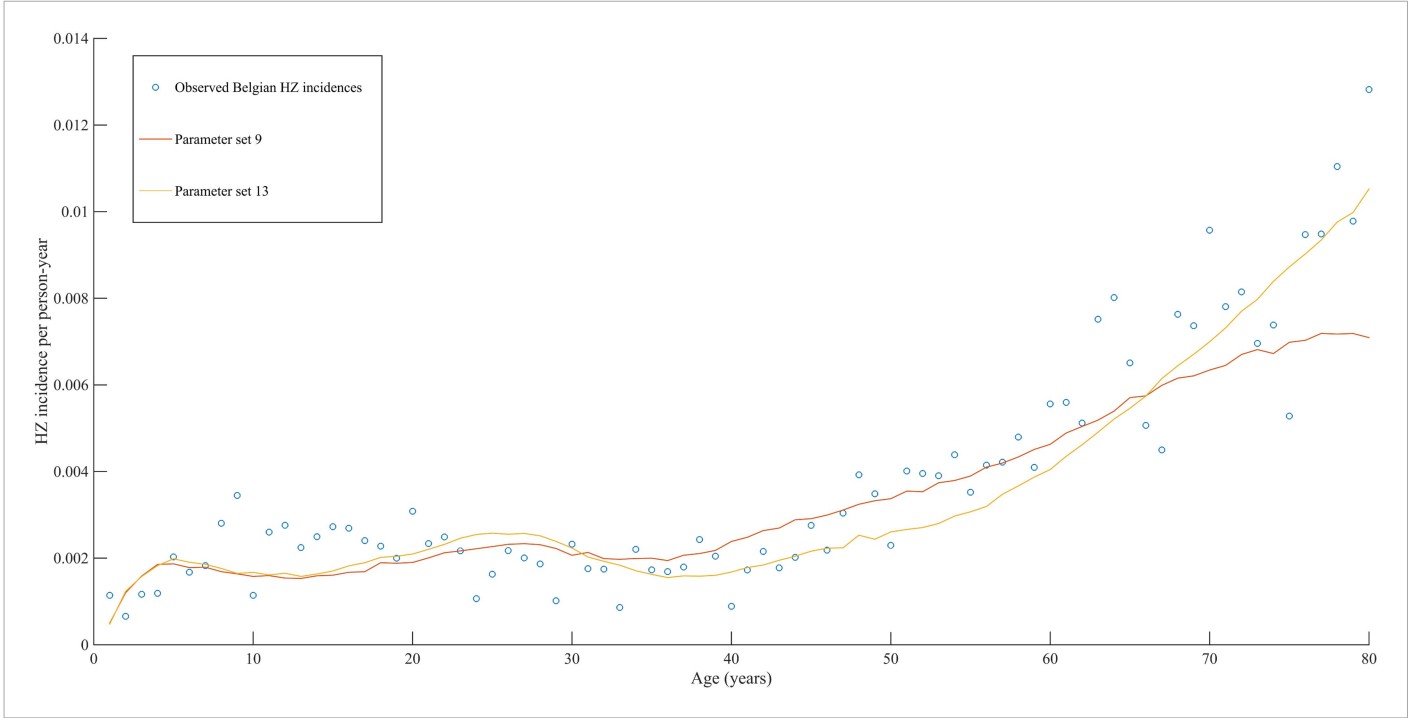

**Figure 2**. Observed (open circles) and simulated (continuous lines) Belgian HZ incidence data by age.

The following figure supplement is available for figure 2:

**Figure supplement 1**. Observed (open circles) Belgian HZ incidence data by age and simulated HZ incidence data (continuous lines) for the 13 best parameter sets with a sensitivity analysis for the HZ infectiousness parameter (values: 0.03, 0.10, 0.17, 0.24, 0.31, 0.38 and 0.45) and three runs per parameter set.

are consistent with those experimentally found in adults re-exposed to VZV by varicella contacts in the household (*Arvin et al., 1983*; *Vossen et al., 2004*) or by vaccination (*Levin et al., 2008*). A possible limitation of our study was that all close contacts with varicella patients were assumed to exert an equal 'average' boosting effect on the exposed individual. Future studies could assess how important it would be to incorporate variability in the impact of an exposure through direct contact with a varicella case, based on characteristics of both the exposed and the infectious person (e.g., age, comorbidity).

The VZV reactivation probability was estimated to be 5% per week. Observed data on VZV reactivation probability are rather sparse and highly divergent regarding study design, sampling site (saliva, blood, cerebrospinal fluid) and results (*Schünemann et al., 1997*; *Mehta et al., 2003*; *Engelmann et al., 2008*; *Birlea et al., 2011*; *Nagel et al., 2011*; *van Velzen et al., 2013*). The observed weekly VZV reactivation probability is a topic of discussion in the literature and varies between 0 and 71%. As such it is difficult to compare our estimates with the range of observed values. We also note that VZV reactivation in our model might only be detectable at the neural ganglia at not (always) necessarily in peripheral tissues. In order for HZ to occur, we assumed that VZV-CMI should be below a relative threshold (following a specific distribution) during VZV reactivation. None of our best fitting parameter sets predicted a significant effect of VZV reactivation on VZV-CMI, implying that endogenous boosting most likely does not have an impact on the occurrence of HZ. This finding is important since the existence of endogenous boosting has been proposed to have an effect in reducing the negative effects of varicella vaccination on HZ incidence. Future experimental studies should focus on confirming our predicted VZV reactivation probability and the lack of endogenous boosting.

The annual decline of VZV-CMI was predicted to be between 1 and 1.5%. This result is lower than the 2.7–3.9% experimentally observed by Levin et al. for individuals older than 60 years (*Levin et al., 2008*). The twofold difference in waning rate can be explained by the explicit disentanglement of waning and

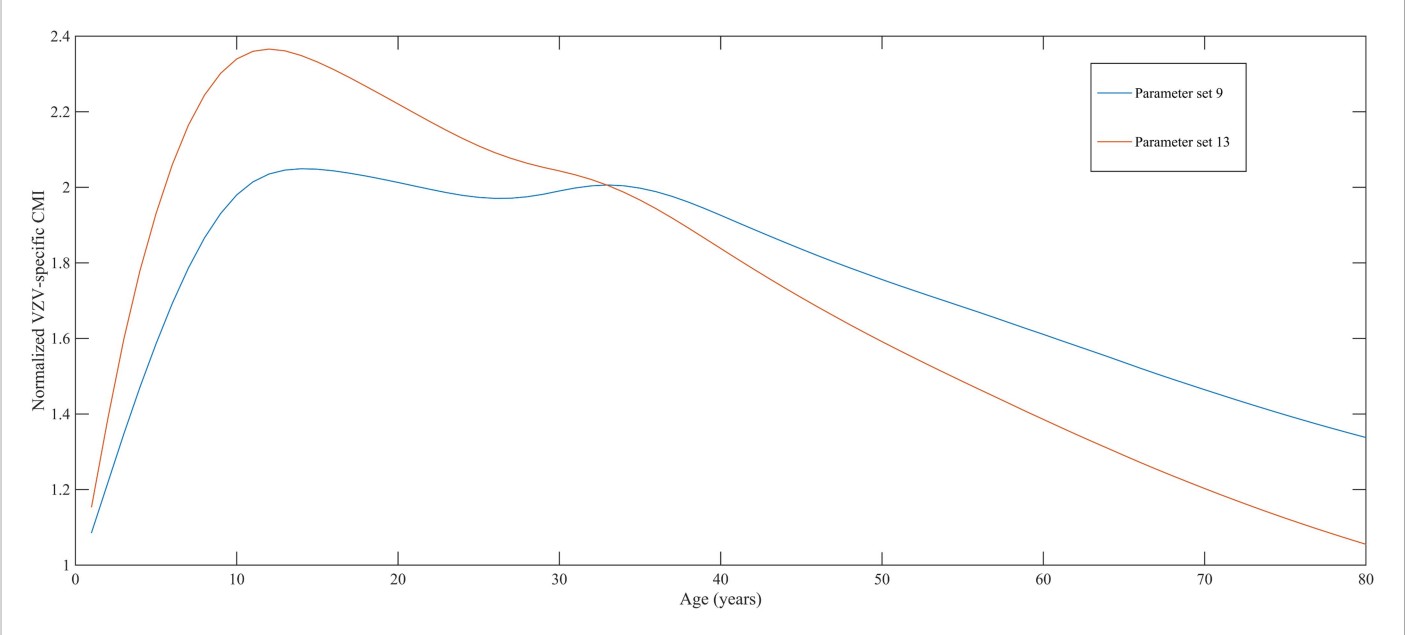

**Figure 3**. Normalized varicella-zoster virus (VZV)-specific CMI averaged over 80 simulation years and over all individuals for the two best parameter sets. Caption: note that this figure shows average dynamics although some individuals will have VZV-specific CMI values below 1 (making them susceptible to HZ).

boosting (with younger age groups having—on average—higher probabilities of being boosted recently thereby actually increasing the observed waning rate). A limitation in our study is the use of a fixed waning rate for all ages. Our results might be interpreted as an averaged result of lower waning rates for younger age groups and higher waning rates for older age groups (as documented by *Levin et al. (2008)*). A higher waning rate in older age groups could for example be caused by chronic CMV infection (*Ogunjimi et al., 2014*). Although different types of waning (both in model specification and age dependency) can be used in this kind of simulation models, we believe that further experimental data documenting VZV specific cellular memory as a function of age is needed so that new waning models can be appropriately formulated. One future avenue of research could be the fitting of our predicted VZV-CMI to observed VZV-CMI data, as this will be a mixture of waning, immunosenescence and boosting. Better VZV-CMI datasets than those currently available are needed to be able to do this. These datasets could and should contain immune responses against different VZV peptides and could differentiate between the different cellular immunity compartments (CD4 vs CD8 and central vs effector memory cells). The use of our VZV IBM could help us identify which VZV-CMI compartment is of importance in controlling HZ.

We used our best fitting models to analyze the effects of a 100% effective varicella vaccine implemented for all 1-year-old children. We predicted a net increase in HZ incidence during 50 years and a 1.75 peak fold increase 31 years after introduction of the vaccination program. This delay in the HZ peak incidence is caused by cohorts born close to the time varicella vaccination was introduced experiencing less repeated boosting instances during their childhood than previously born cohorts (a mechanism that is similar to the progressive immunity model proposed in the deterministic VZV model by *Guzzetta et al. (2013)*). Although increases in HZ incidences following universal childhood varicella vaccination have been noticed, some authors have attributed these increases to a background evolution that was already present prior to CP vaccination (see discussion in *Ogunjimi et al. (2013)*). However, our analysis shows that proper documentation of significant increases in HZ incidence might not be possible during the first 10 years, even if a 100% effective vaccine would be used. For instance, during the first 10 years of the US program, both uptake and efficacy with the initial single dose vaccination programme were far below 100%. Although our model predicts a much lower duration of boosting than used hitherto in compartmental models (*Ogunjimi et al., 2013*), some of our overall HZ

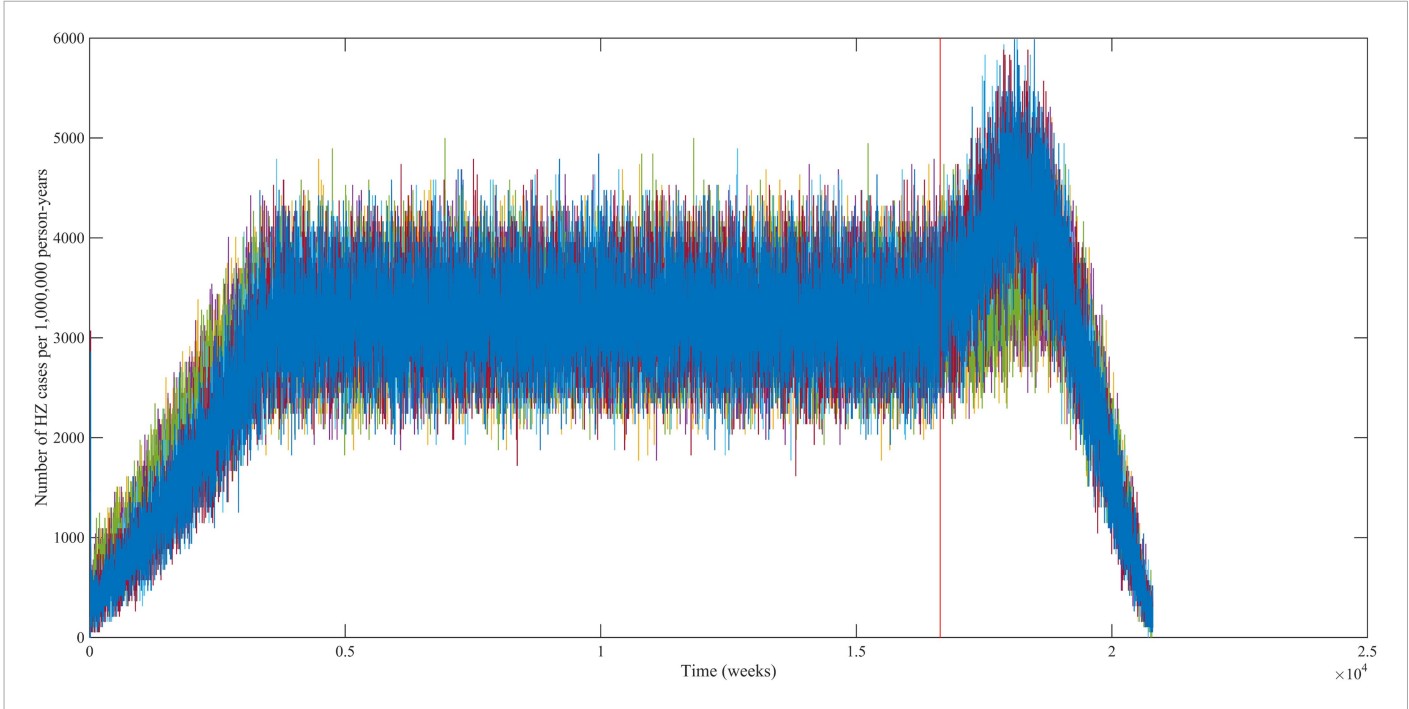

**Figure 4**. Predicted HZ incidence (aggregated for all ages) over time with a CP vaccine for 1 year olds using the best-fitting parameter sets. The red line indicates the moment of CP vaccine introduction, which is assumed to be 100% effective.

projections are qualitatively similar. However, in contrast to earlier model estimates our VZV IBM predicts that 31–40 year olds contribute the most to the peak in HZ incidence following varicella vaccination. Some observational studies found no effect of varicella vaccination on HZ incidence for those aged 60 years and older (*Hales et al., 2013*). This could be compatible with an overall HZ incidence increase due to rising HZ incidence among 31–40 year olds. Importantly, younger adults have been shown to be less likely to develop post-herpetic neuralgia (*Opstelten et al., 2002*; *Drolet et al., 2010*). Thus although our aggregated predictions regarding the increase in HZ incidence following varicella vaccination may appear similar to those published using deterministic models, cost-effectiveness analyses using our VZV IBM would be more in favor of universal childhood varicella vaccination.

A major benefit of our modeling approach is the possibility to verify our best–fit parameter values via experimental studies, or vice versa, to adapt parameter values when empiricism delivers new insights. For example, if a new experimental study would prove the existence of endogenous boosting, this could be readily implemented in our VZV IBM so that parameter sets could be estimated, conditional on the existence of endogenous boosting. Although our IBM has given us valuable insights into between-host and within-host VZV dynamics, other influential factors could be introduced in future versions. These factors could relate to CMV-immunosenescence (*Ogunjimi et al., 2014*), maternal antibodies, reduced VZV-CMI induction if infected during the first year of life as this might improve the prediction of the teenage group (*Terada et al., 1994*), co-infection with other viruses (*Ogunjimi et al., 2015*), risk groups (for e.g., immunosuppressed individuals) and HLA types (*Meysman et al., 2015*). Although current deterministic compartmental models should be able to partially account for these effects, a VZV IBM seems better suited to directly model the influence of these immunity-perturbing factors. Future VZV IBM should also explore more realistic vaccination scenarios as well as inter-country variability (*Poletti et al., 2013*).

We conclude that our VZV individual-based model has explicitly estimated the duration of exogenous boosting to be limited to only 1 or 2 years and that there was no significant effect from endogenous boosting.

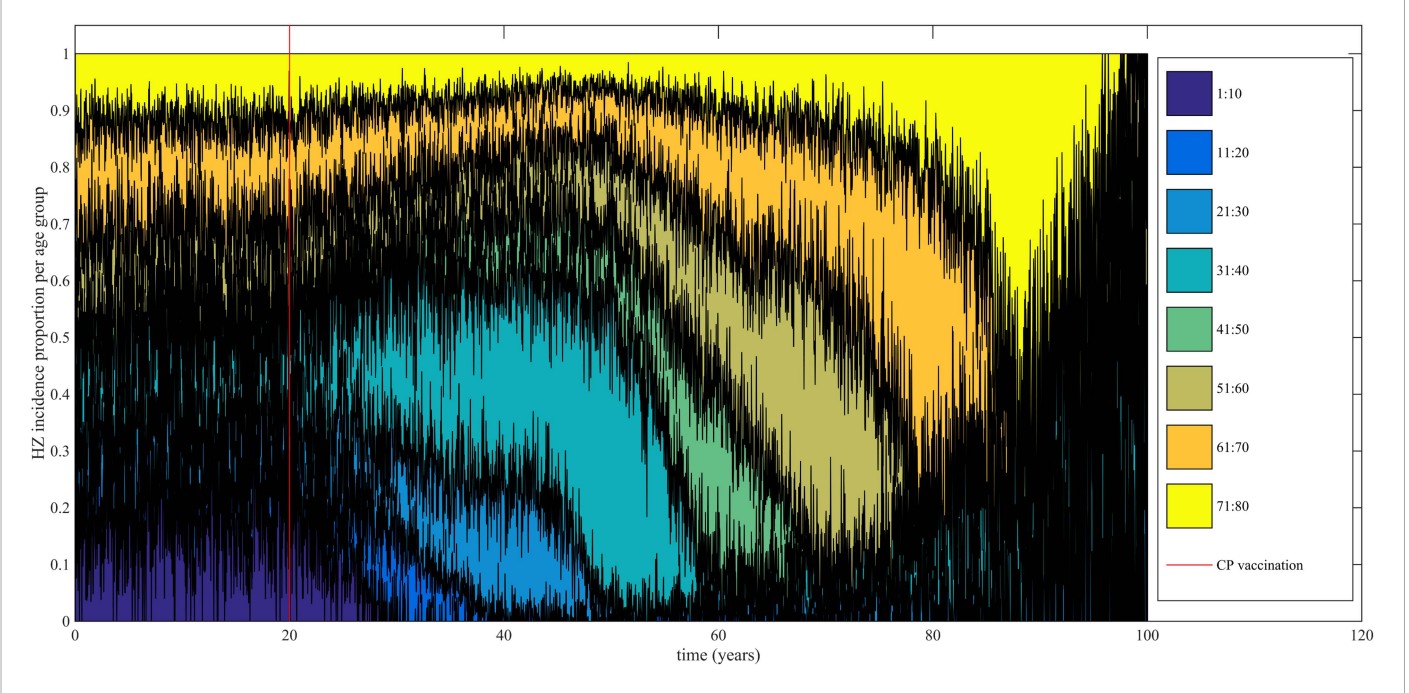

**Figure 5**. Time-evolution of the relative contribution to HZ incidence per age group before and after introduction of 100% effective varicella vaccination for 1 year olds.

## Materials and methods

### Model overview

We present an individual-based model in which the individual's risk of HZ is determined by the individual's VZV-CMI vs the so defined 'Force of Reactivation (FoR)' that represents the strength of VZV reactivation (as detailed in the Modeling VZV reactivation paragraph). In contrast to classical deterministic epidemiological models, our model is not explicitly compartmentalized by means of ad hoc defined epidemiological groups. Instead the main 'flow diagram' represents the evolution of VZV-CMI with time and under several events (for more details see the next paragraphs). Briefly, *Figure 6* illustrates that individuals are born susceptible and that VZV-CMI is equal to zero during this time period. After exposure to chickenpox or HZ, VZV-CMI receives an initial value. Next, VZV-CMI is expected to undergo waning and several events can occur. The first event depicted (but the reader should realize that the timing of the events are interchangeable) is exogenous boosting after re-exposure to either chickenpox or HZ. Exogenous boosting is assumed to increase VZV-CMI temporary, after which decay towards 'the original trajectory of VZV-CMI' is assumed to occur. Next, we depicted unsuccessful reactivation of VZV that is assumed to cause endogenous boosting of VZV-CMI (again followed by a decay). However, if the FoR is relatively higher than VZV-CMI, VZV reactivation will be successful and will cause HZ (followed by a reinstatement of VZV-CMI).

### Demographics

The dynamics of the synthetic population of the individual-based model are based on Belgian population and mortality data normalized to a fixed total population of 998,400 individuals (Eurostat, 2012). The chosen population size is the result of a trade-off between ensuring sufficient heterogeneity and a manageable computational burden. Natural deaths are selected based on the age-dependent mortality rate. Per time step, the number of newborns equals the number of deaths to obtain a constant population size. To conduct predictions over many years, a stationary demographic structure is required throughout the simulated period. The Belgian population from 2012 has an

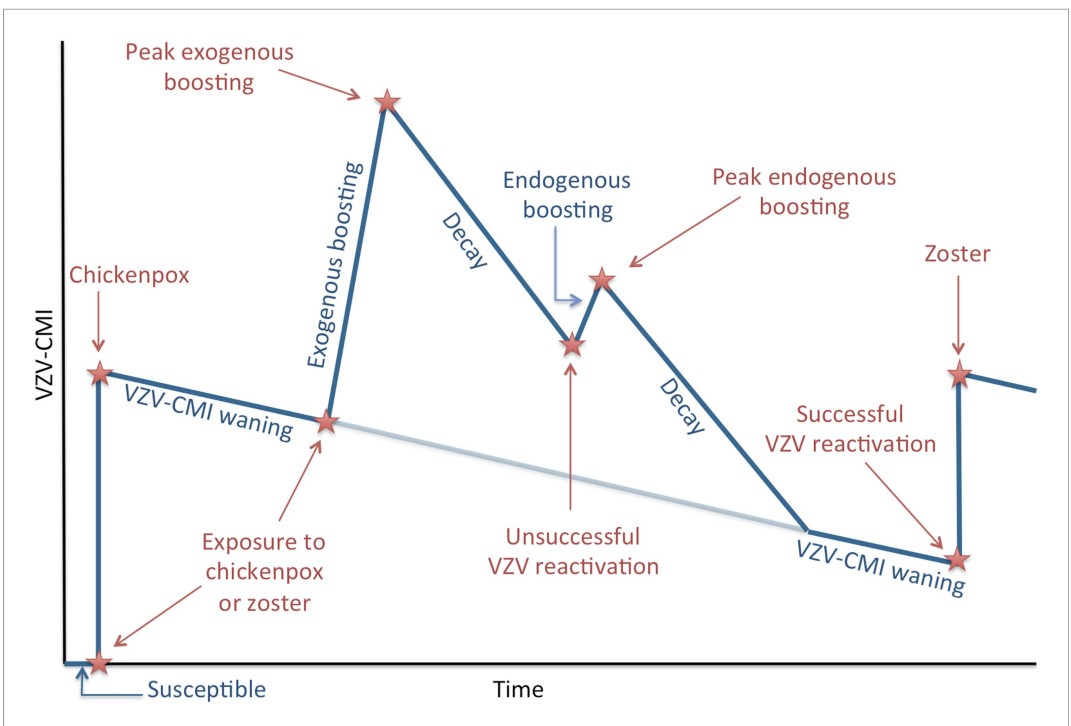

**Figure 6**. Simplified dynamics of VZV-CMI, VZV reactivation and boosting events as modeled. The sequence of exogenous boosting and VZV reactivation can be switched.

overrepresentation of people from age 40 to 60 years (see *Figure 7*). Applying a fixed age-dependent mortality rate in combination with the replacement by newborns would results in an oscillating population distribution over time with respect to age. Therefore, the initial age distribution is not based on the Belgian population count from 2012 but on the survivor function estimated from data on the number of deaths per capita. The survivor function by age is estimated from data on the number of deaths per capita by $m(a) = \exp(-\int_0^a \mu(t)dt)$ with age-dependent mortality rate $\mu$ using a thin plate regression spline model with the *gam*-function in the R-package *mgvc* (*Wood, 2011*; *Hens, 2012*). *Figure 7* illustrates the survivor function by age together with the Belgian population and mortality rates in 2012.

The model runs in time steps of 1 week and people with week-age 53 move to the next age class. To obtain homogeneous age transitions throughout the simulated period, initial week-ages are randomly assigned between 1 and 52. People from the last age class with 53 weeks are removed from the population so that the demographic structure remains stable throughout the simulated period.

### Modeling dynamics of primary VZV infection

At the start of the simulation, 30 individuals between ages 1 and 3 are randomly infected with CP. The weekly probability $\lambda_{i,t}$ for a susceptible individual to become infected with VZV (after contact with at least one infectious individual) is calculated by $\lambda_{i,t} = 1 - \prod_{n=1}^{80} (1 - w(i,n))^{\sum CPI_n} \cdot (1 - m \cdot w(i,n))^{\sum HZI_n}$ with $w(i,n)$ the weekly effective contact probability for an individual from group $i$ with a random individual from group $n$, $m$ the relative HZ infectiousness (empirically estimated to be 0.17 [cf. paragraph 'Modeling VZV endogenous reactivation']) and $CPI_n$ and $HZI_n$ the total number of infectious individuals per age class for CP and HZ, respectively. This formula is thus constructed by the complement of the probability that an individual did not have a successful contact with any of the chickenpox or HZ patients. The VZV infection probability, $w(i, n)$, is based on empirical social contact data as described elsewhere (*Ogunjimi et al., 2009*). Here $w(i, n)$ equals the number of close contacts per week lasting longer than 15 min between two random individuals in age classes $i$ and $n$ multiplied by the best fitting proportionality parameter q = 0.181 based on Belgian social contact and VZV seroprevalence data (*Ogunjimi et al., 2009*).

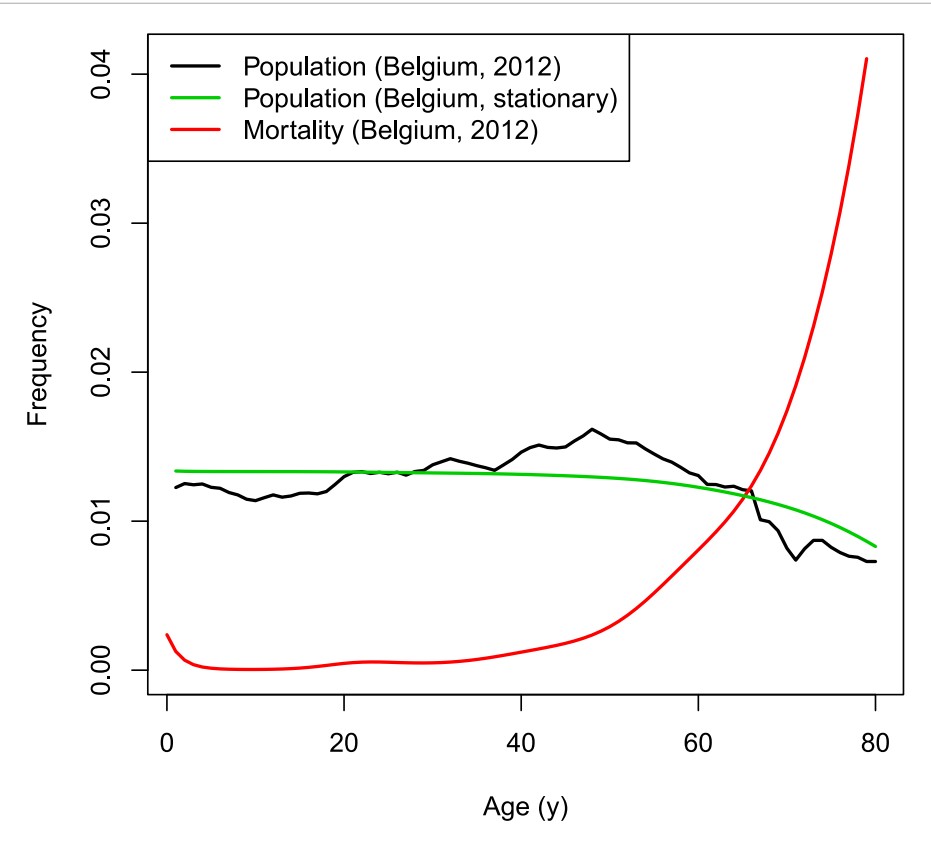

**Figure 7**. VZV IBM demography.

Individuals infected for the first time with VZV are infectious for 1 week after an incubation period of 2 weeks. Next, they are CP recovered and receive a normalized CMI value of 1 ± a randomly distributed factor (normal distribution with variance 0.1 as suggested by *Terada et al. (1994)*).

## Modeling waning of VZV-CMI

Once arrived in the CP recovered state, VZV-CMI starts waning at a weekly rate via the multiplication with (1—waning-rate). The waning-rate (see *Table 2*) is informed by the annual decline of 2.7–3.9% per age-year as noted by baseline VZV-CMI values by *Levin et al. (2008)*. In all model steps, waning is applied to all variables. Note that in our model waning and ageing are indistinguishable.

## Modeling exogenous re-exposure to VZV

At each weekly update, the CP recovered individuals have a probability $\lambda_{i,\,t}$ to receive a boost of VZV-specific immunity. Although HZ is less infectious than CP, we assume that if boosting has occurred, the magnitude of VZV-CMI boosting will be similar for both CP and HZ. To analyze the effect of boosting, we retain a CMI value for each individual with and without boosting.

In the first 6 weeks after exogenous boosting, VZV-CMI is assumed to increase linearly up to 'Peak fold increase' times the pre-boosting value (*Levin et al., 2008*), but is limited to a maximum of 4 times the VZV-CMI value without re-exposure. The 6 week duration between the boosting event and the peak has been influenced by the *Levin et al. (2008)* data. Limiting the effect of boosting to a factor 4 is based on the recent finding that pediatricians, highly exposed to CP, have T-cell values that are on average 3–4 times higher than controls, but not higher (*Ogunjimi et al., 2014*).

Next, VZV-CMI will decrease following one of three different boosting scenarios (as shown in *Figure 8*) until it reaches the pre-boosting value again (after a 'duration of boosting'):

**Table 2**. Initial parameter sets

| Parameters | Step 1 | Step 2 |
|---|---|---|
| Annual waning rate (%) | 2.0 | 0.5 |
|  | 3.0 | 1.0 |
|  | 4.0 | 1.5 |
|  | – | 2.0 |
|  | – | 2.5 |
| Boosting scenario | 1 | 3 |
|  | 2 | – |
|  | 3 | – |
| Duration of boosting (years) | 1 | 1 |
|  | 2 | 2 |
|  | 4 | 3 |
|  | 7 | 4 |
|  | 12 | 5 |
|  | – | 7 |
|  | – | 10 |
|  | – | 12 |
|  | – | 15 |
| Peak fold increase following exogenous boosting | 1 | 1.3 |
|  | 1.6 | 1.6 |
|  | 2.2 | 1.9 |
|  | – | 2.2 |
|  | – | 2.5 |
| VZV weekly reactivation probability (%) | 0.01 | 0.001 |
|  | 0.1 | 0.05 |
|  | 0.3 | 0.01 |
|  | 0.5 | 0.015 |
|  | – | 0.1 |
|  | – | 0.2 |
|  | – | 0.3 |
|  | – | 0.4 |
| Distribution threshold VZV-CMI for HZ | 1 | 1 |
|  | 2 | 2 |
|  | 3 | 4 |
|  | 4 | – |
| Peak fold increase following endogenous boosting | 1 | 1 |
|  | 1.4 | 1.2 |
|  | 1.8 | – |
|  | 2.2 | – |

1. Exponential decrease from peak to pre-boosting value based on VZV ELISPOT CMI data after Zostavax vaccination as published by *Levin et al. (2008)*. *Levin et al. (2008)* presented ELISPOT data on Zostavax vs placebo-vaccinated individuals and the relative increases were (visual reference to figure in the Levin et al. paper): 120% (6 weeks), 60% (1 year), 50% (2 years) and 40% (3 years). Using these average values we performed a regression analysis to estimate the VZV-CMI exponential decay rate per week.

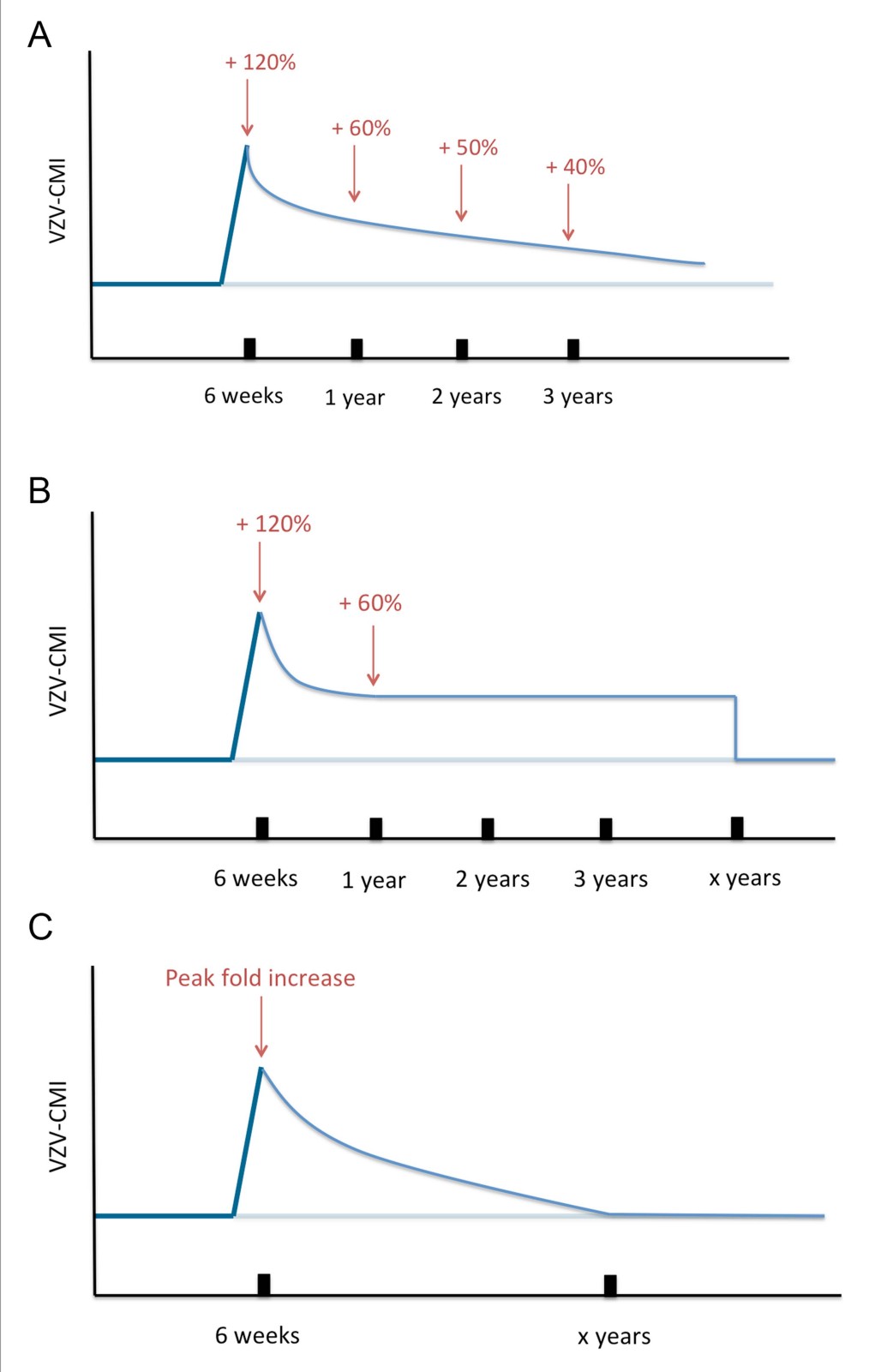

**Figure 8**. Three different boosting scenarios. (**A**) Illustrates the exponential decline parameterized by a peak (+120%) at 6 weeks, (+60%) 1 year later, (50%) 2 years later and (+40%) 3 years later as presented by the Zostavax vaccine trial by Levin et al. (**B**) Illustrates the exponential decline from peak (+120%) to (+60%) 1 year later and constant for x years (as defined by the parameter set) after wards, as a modified interpretation of the

*Figure 8. continued on next page*

*Figure 8. Continued*

results of the Zostavax vaccine trial by Levin et al. (**C**) Illustrates the increase to a peak value as defined by the parameter set that is followed by an exponential decline so that the pre-boosting value is reached after x years.

2. Exponential decrease from peak to 60% of pre-boosting value ([*Levin et al., 2008*], based on simulation from peak to year 1, cf. supra), followed by a constant value for x years with x defined by the parameter set, but ≥3 years. After x years, VZV-CMI returns to the pre-boosting value. This scenario is compatible with the assumption that memory cells have a predefined and fixed lifespan as implied by *Westera et al. (2013)*.

3. Exponential decrease from peak to pre-boosting value based on the boosting period of x years as defined by the parameter set. The exponential decay is defined by the peak and duration of boosting: $Y(t) = P \cdot Y_0 \cdot e^{-\frac{\ln(P)}{x} \cdot t}$ with Y(t) = VZV-CMI (disregarding age), P = peak fold increase, $Y_0$ = VZV-CMI prior to entering boosting sequence (= 'original VZV-CMI') and x = duration of boosting. This function is constructed by the boundary conditions that Y(t = x) = $Y_0$ and Y(t = 0) = P*$Y_0$.

It is important to clarify that if a new boosting event occurs during an ongoing boosting sequence, the VZV-CMI value attained right before the new boosting event occurs, is assumed to be the baseline 'pre-boosting' reference for the new boosting sequence. This means that for scenario 3 in case of a new boosting event 6 weeks (and mutatis mutandis for the other situations) after the first boost VZV-CMI evolves as

$$Y(t) = P \cdot Y_1 \cdot e^{-\frac{\ln(P)}{x} \cdot (t-t_1)} = P \cdot \underbrace{P \cdot Y_0 \cdot e^{-\frac{\ln(P)}{x} \cdot t_1}}_{Y_1} \cdot e^{-\frac{\ln(P)}{x} \cdot (t-t_1)} = P^2 \cdot Y_0 \cdot e^{-\frac{\ln(P)}{x} \cdot t},$$

with the subscript 1 referring to the situation when the second boosting event occurs.

This shows that boosting during an ongoing boosting episode prolongs the time before the 'original VZV-CMI (Y(t) = $Y_0$)' is reached again.

## Modeling VZV reactivation

After primary infection, VZV is assumed to remain latent, but capable of reactivation. The frequencies of VZV reactivation used in the parameter sets were informed by observed VZV reactivation frequencies in random samples from healthy individuals (2% in blood [*Schünemann et al., 1997*]; 0 out of 112 saliva samples [*Mehta et al., 2003*]; 2.5% in saliva [*Nagel et al., 2011*]), immunosuppressed patients (8.1% from various sites [*Engelmann et al., 2008*]), individuals with malignancies (7.5% in blood [*Malavige et al., 2010*]) and HIV patients (9% in saliva [*van Velzen et al., 2013*]; 16% in cerebrospinal fluid [*Birlea et al., 2011*]). The consequence of reactivation can either be endogenous boosting or clinical reactivation (HZ) and this is defined by the difference between VZV-CMI and the FoR.

The FoR defines the VZV-CMI needed to resist clinical reactivation and if VZV-CMI < FoR, reactivation will lead to HZ. The reader should imagine the FoR to represent independent reactivation behavior of VZV and whether this reactivation will lead to HZ or endogenous boosting will depend on the value of VZV-CMI. The FoR deviates per time step and individual by means of a gamma probability density function. This gamma function is chosen as it represents the summation of unknown biological phenomena that are assumed to have an exponential distribution. The parameter set includes four different gamma distributions (see *Table 2* and *Figure 9*). HZ individuals are assumed to be infectious for 1 week and receive a VZV-CMI reset to 1 ± random factor (normally distributed, cf. primary infection). There exists no data on the relative infectiousness of HZ patients nor is it likely that this can be estimated through experiments. We chose to approximate the relative infectiousness of HZ patients by the relative infectiousness of breakthrough CP patients (*Bernstein et al., 1993*). This has previously been estimated at 0.17 (*Brisson et al., 2002*). Although HZ infectiousness was needed to maintain circulation of VZV in our model, the relative effects on overall VZV transmission are—compared to CP—marginal. *Figure 2—figure supplement 1* shows the results of a sensitivity

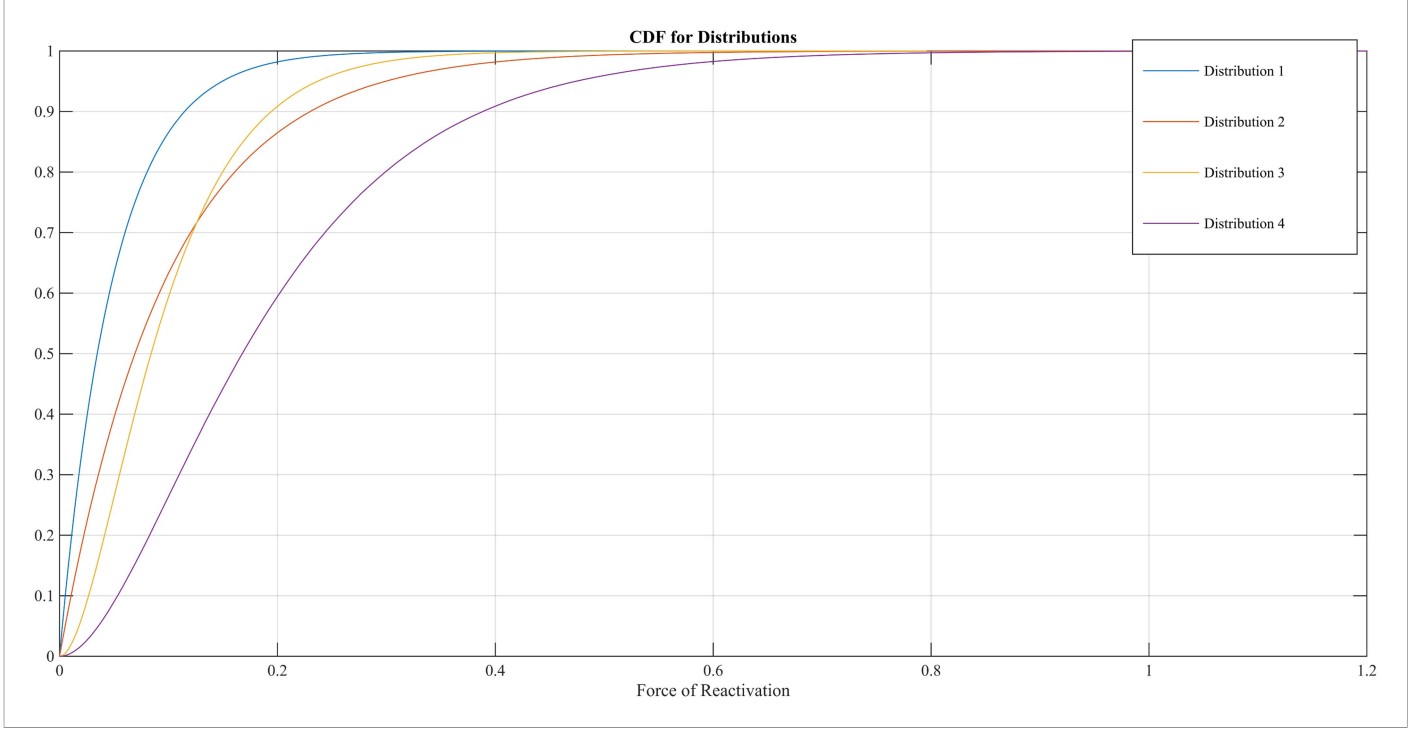

**Figure 9**. Different cumulative distribution functions (CDF) for Force of Reactivation (FoR).

analysis in which we varied HZ infectiousness from 0.03 to 0.45 for the 13 best fitting parameter. As can be seen in *Figure 2—figure supplement 1* our results are quite robust. Reinstallation of VZV-specific immunity after HZ is based on experimental data showing that a second case of HZ only occurs in about 5% of individuals (*Yawn et al., 2011*) and that VZV-CMI is higher in recovered HZ patients than in age-matched controls (*Weinberg et al., 2009*).

**Table 3**. Step 2 parameter set selection

| Parameters | Best parameter sets + deviance +5% | Most prevalent parameters in Q2.5 |
|---|---|---|
| Annual waning rate (%) | 2.0 | 2.0 |
| Boosting scenario | 3 | 3 |
| Duration of exogenous boosting (years) | 1 | 1 |
| | 4 | 2 |
| | – | 4 |
| | – | 7 |
| | – | 12 |
| Peak fold increase following exogenous boosting | 1.6 | 1.6 |
| | – | 2.2 |
| VZV weekly reactivation probability (%) | 0.01 | 0.01 |
| | 0.1 | 0.3 |
| Distribution threshold VZV-CMI for HZ | 2 | 1 |
| | 4 | 2 |
| Peak fold increase following endogenous boosting | 1 | 1 |

If VZV-CMI ≥ FoR, endogenous boosting occurs followed by an exponential decrease according to one of the scenarios described in the previous section. The peak following endogenous boosting, however, is restricted to be at most equal to the peak following exogenous boosting. In addition, we assume that an endogenous boosting sequence will always be overruled by a new successful exogenous re-exposure boosting episode.

## Statistical and computational details

Simulations were performed using Matlab 2012b on the Flemish VSC supercomputer. Simulations ran for 320 years and model output was obtained by averaging the age-specific results over the last 80 years. The main outputs were CP incidence, HZ incidence, VZV serology and VZV-CMI. In order to optimize the fitting procedure, we performed a two-step parameter set analysis. The following 7 parameters were estimated by means of the fitting procedure: VZV-CMI waning rate, type of boosting scenario, duration of exogenous boosting, peak fold increase following exogenous boosting, VZV reactivation probability, FoR distribution and the peak fold increase following endogenous boosting.

In the first step we ran each parameter set three times. We calculated per set the Binomial likelihood by fitting the HZ age-specific output data to Belgian HZ incidence data (*Bilcke et al., 2012*). Next, we selected the parameter set leading to the lowest mean deviance (= −2*loglikelihood) based on the three repetitions with different stochastic seed and all other parameter sets with deviance +5% at most in order to account for model selection uncertainty (*Castro Sanchez et al., 2013*). In order to broaden the parameter selection, we also selected the most prevalent (marginal) parameter values in the lowest mean deviance 2.5% percentile (see *Table 3*).

In the second step we adapted our parameter ranges and intervals according to the best fitting values to obtain new parameter combinations (see *Table 3*). Again, we ran each parameter set three times and calculated the mean deviance. The best parameter sets were defined by those parameter sets that had at least one run with a deviance within the 5% range of the deviance of the best fitting parameter set.

Given the fact that some selected parameter values were on an unexplored border of the parameter grid, we studied whether more extreme values for the border parameters led to better results (and continuing if deviance was within the second step best deviance +5%). Doing this, we allowed the other parameters to vary for one unit in both parameter directions.

## Predicting the effects of CP vaccination on Belgian HZ incidence

We used our best parameter sets and introduced a simplistic hypothetical single dose 100% effective CP vaccine (without waning of vaccine-induced immunity) for all children ageing between 1 and 2 years. Vaccinated individuals were assumed not to be susceptible to HZ.

## Acknowledgements

The computational resources and services used in this work were provided by the VSC (Flemish Supercomputer Center), funded by the Hercules Foundation and the Flemish Government—department EWI. In particular, we thank Drs Bex, Becuwe and Backeljauw for their support in using the Supercomputers.

## Additional information

### Funding

| Funder | Grant reference | Author |
|---|---|---|
| Fonds Wetenschappelijk Onderzoek | G040912N | Benson Ogunjimi |
| University of Antwerp | Special Research Fund predoctoral fellowship | Lander Willem |

The funders had no role in study design, data collection and interpretation, or the decision to submit the work for publication.

### Author contributions

BO, LW, PB, NH, Conception and design, Acquisition of data, Analysis and interpretation of data, Drafting or revising the article

**Author ORCIDs**
Benson Ogunjimi, http://orcid.org/0000-0002-0831-2063

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
