## [Decision Letter]

[Editors’ note: this article was originally rejected after discussions between the reviewers, but the submission was accepted after an appeal against the decision.]

Thank you for choosing to send your work entitled “Integrating between-host transmission and within-host immunity to analyze the impact of varicella vaccination on zoster” for consideration at *eLife*. Your full submission has been evaluated by Prabhat Jha (Senior editor) and three peer reviewers, one of whom is a member of our Board of Reviewing Editors, and the decision was reached after discussions between the reviewers. Based on our discussions and the individual reviews below, we regret to inform you that your work will not be considered further for publication in *eLife*.

The following individuals responsible for the peer review of your submission have agreed to reveal their identity: Mark Jit (Reviewing editor); Alessia Melegaro (reviewer #3). Another reviewer chose to remain anonymous.

All of us agreed that it is innovative and addresses an important issue which is being explicitly modelled for the first time, i.e. the role of the dynamics of cell-mediated infection and endogenous boosting in VZV reactivation. Given that this concern has had a large influence on vaccine decision making in Europe and elsewhere, the topic is of considerable public health importance. However, we had a number of major concerns with the paper which would require substantial changes to the methodology to address.

Below are the most substantive concerns; however we would encourage the authors to look at the more detailed comments from each reviewer as well.

1) The model is itself is hard to understand and therefore to assess what it adds to the existing literature. The Methods section plunges straight into a discussion of the estimation of the various parameters. It is not clear whether the follows the traditional Brisson/Edmunds/van Hoek approach, the Guzzetta progressive immunity one (AJE 2013) or a different approach entirely. It would be helpful to start with an overview of the model structure (with a written description and/or compartmental flow diagram), as well as the main ways it differs from previous models.

2) It is not clear what the conclusions add to existing modelling work on the same topic. The manuscript criticises existing deterministic models used in current research in the field. However, the main conclusions of the paper (e.g. with regards to vaccine impact) are in line with those of available deterministic models. Although the rise in herpes zoster is predicted to occur earlier in life, it is unclear whether the magnitude of the HZ rise would be smaller, the same or larger than in previous models, and hence the overall significance of this age shift. The fact that a perfect vaccine is used makes the comparison particularly difficult, although it is understandable why this was done in order to show effects more clearly. Hence it would be important to explain clearly what the gains achieved by an individual-based framework taking into account cell-mediated infection explicitly are.

3) In addition, the manuscript points out that a variety of models are able to fit age-specific incidence or prevalence data so conclusions may be a function of model structure rather than biological fact. However, the model as it stands has at least 6 free parameters (i.e. more than in currently used deterministic models), which appear to be fitted to a single output, a zoster age-specific incidence profile. This should worse over-fitting problems, thereby increasing rather than reducing the lack of identifiability. There needs to be some way to address this problem, or at least more extensive sensitivity analyses to show that the conclusions are robust to the range of possible parameter values.

4) Furthermore, like previous models, the conclusions are not completely independent of subjective model choices, since the form and sometimes the parameter values of functions representing cell-mediated immunity and boosting have to be pre-determined (albeit with reference to empirical work). In particular, in the three exogenous boosting scenarios, a number of parameters are simply assumed and no explanations on their values are provided. For example the force of boosting (subsection “Modeling exogenous re-exposure to VZV”) is assumed to occur at the same rate as the force of infection. This assumption (which contrasts with the earlier remark that the actual magnitude of the boosting effects is unknown) should be emphasized, as it might be a main responsible of the fact that seven of the best models “include exogenous boosting” (see Results). Better justification for all the assumptions and assumed parameter values is required; at present it is not clear that this is the “final answer” to the exogenous boosting question.

Reviewer #1:

This paper deals with an important issue, i.e. the extent to which childhood varicella vaccination may cause an increase in HZ in adults due to reduction in exogenous boosting. The observational and ecological evidence about boosting are hard to interpret precisely. As the authors point out, a variety of mathematical models are able to fit age-specific incidence or prevalence data so conclusions may be a function of model structure rather than biological fact. This paper offers a number of improvements that may enhance the believability of conclusions, particularly an individual-based structure which allows tracking of within-host immune dynamics, allowing the duration of exogenous boosting to be fitted rather than largely assumed.

The main difficulty is that the model itself is hard to understand and therefore to assess what it adds to the existing literature. The Methods section plunges straight into a discussion of the estimation of the various parameters. It would be helpful to start with an overview of the model (with a written description and/or compartmental flow diagram), as well as the main ways it differs from previous models.

The comparison with previous models (which have been influential to vaccine policy in much of Europe) needs to be a bit sharper. In particular, it is not clear whether the magnitude of the HZ rise is smaller, the same or larger than previous models (e.g. Brisson). The fact that a perfect vaccine is used makes the comparison particularly difficult, although it is understandable why this was done in order to show effects more clearly. Hence even though the rise in HZ occurs earlier in life, it is not clear what the impact on cost-effectiveness will be.

Also it should be noted that the conclusions are not completely independent of subjective model choices, since the form and sometimes the parameter values of functions representing cell-mediated immunity and boosting have to be pre-determined (albeit with reference to empirical work), as is inevitable. Hence it is not clear that this is the “final answer” to the exogenous boosting question.

However, the model clearly adds a great deal of additional insight to the existing debate, including the use of data that was not previously used in this fashion.

Reviewer #2:

The paper has some innovative aspects, as it represents a first attempt (to my knowledge) to explicitly include into a single model for VZV transmission dynamics and reactivation (and vaccination), the trend in cell mediated immunity (CMI) and endogenous boosting. As trends in CMI and the role of boosting are thought to be the primary determinants of VZV reactivation, explicit consideration of such effects (instead than e.g. implicitly as in available models) is worthwhile.

I have however some major points that should be carefully tackled. Authors criticize deterministic models used in current research in the field (Introduction, “One of the main points of criticism ... equilibrium states.”) Since however some of the main conclusions of this paper are in line with those of available deterministic models (e.g. in regards of the impact of vaccination) authors should explain clearly which are the possible gains allowed by IBMs, particularly in what respects the present approach should be superior to the deterministic models used in the literature. My main point here deals with identifiability of model parameters. The current model has at least 6 free parameters (i.e. more than in the currently used deterministic models), which as I understand are fitted to a single output, a zoster age-specific incidence profile. This should worse over-fitting problems, thereby emphasizing rather than reducing, the aforementioned criticism.

I am not entirely clear about the three hypotheses about CMI decrease after a boosting event (subsection “Modeling exogenous re-exposure to VZV”). In subsection “Modeling waning of VZV-CMI” a hypothesis has been made about CMI decline after primary CP infection. I believe one should take this – for reasons of both modeling parsimony and simplicity – as a baseline holding also for CMI decline after a boosting event, especially given that the data by Levin et al. used by the authors to parametrize the model did not seem to refer to CMI decline after primary chickenpox only. Other hypotheses could then naturally be grounded against this baseline. In particular I am not clear about the meaning of HP1, in subsection “Modeling exogenous re-exposure to VZV”, “Exponential decrease from peak to pre-boosting value based”. Why do you want to prevent that decline continues beyond the pre-boosting situation? The Hp3 formula is incomprehensible and needs careful explanation.

The force of boosting (subsection “Modeling exogenous re-exposure to VZV”) is assumed to occur at the same rate as the FOI. This assumption (which contrasts the authors' remark that the actual magnitude of the boosting effects is unknown) should be emphasized, as it might be a main responsible of the fact that seven of the best models “include exogenous boosting” (see Results).

About the modeling of the Force of reactivation (subsection “Modeling VZV endogenous reactivation”), recent papers have modeled it as an increasing function of age (=senescence), and of time elapsed since last boosting episode. In the paper this is modeled as a purely random factor “The FoR deviates per time step and individual by means of a gamma function.” Am I understanding correctly? In my opinion that should be taken as a (useful) baseline against which, however, to ground alternative, possibly more realistic hypotheses. Therefore this point should be motivated more carefully. How was this gamma (density?) parametrized? This is important because one of the main results on the impact of vaccination i.e. the young age of zoster cases in the post-vaccination period (“Noteworthy is the relative dominance of the 31–40 year...”) might strongly follow from this hypothesis.

Reviewer #3:

This work aims at answering some relevant questions about VZV infection and in particular estimating parameters that describe the exogenous and endogenous boosting effect, a key aspect when considering the introduction of varicella vaccination. For this reason I believe that the paper, in principle, is very relevant and of interest for the research community that is involved in VZV modeling and policy decision making.

However, also considering the standard of the *eLife* journal, I believe the paper cannot be accepted in its current form. Some major improvements and changes are required to make it fit for publication. In particular, specific modeling assumptions and fitting procedures need to be carefully re-addressed and properly clarified before consideration for publication. See the list below for the required amendments:

1) The analysis is based on an Agent-Based Modeling framework. These models require a very thorough validation procedure, especially when considering a realistic demographic population. The authors do not show any model output that convinces us that the demographic component is suitable to model VZV in Belgium. I believe that these validations outputs should be presented.

2) Details of the model structure used are missing. Is the structure following the traditional Brisson/Edmunds/van Hoek approach or the Guzzetta progressive immunity one (AJE 2013)? Details on this should be available to the reader.

3) The three exogenous boosting scenarios: a number of parameters are simply assumed and no explanations on their values are provided. In particular, why six weeks to reach the peak after exogenous boosting? Also, in scenario 2, 60% of pre-boosting value (why?), steady state for x years (?) and why >= 3 years? A better explanation on these parameter values is required. Also what is the biological explanation for the return to pre-boosting values after x years? Finally, in scenario 3 there's a little bit of confusion between delta t and x. The authors should check the formula of Y(t).

4) The lines “HZ individuals are assumed to be infectious for 1 week and receive a VZV-CMI reset to 1 +/− random factor (normally distributed, cf. primo infection) and “HZ only occurs in about 5% of individuals [40] and that VZV-CMI is higher in recovered HZ patients than in age-matched controls” seem contradicting. What is the post HZ value of VZV CMI? As after primary infection or higher?

5) The sentence “Although HZ infectiousness was needed to maintain circulation of VZV in our model, the relative effects on overall VZV transmission are – compared to CP – marginal”: it is not clear here why you need HZ infectiousness to maintain circulation of VZV infection. If it is post-vaccination we don't care to maintain VZV circulation. If it is pre-vaccination, then there is something wrong if the model need HZ to keep VZV infection.

6) The approximation of HZ infectiousness to varicella breakthrough is a strong assumption. at the least we need to check the sensitivity of model results to variations of this parameter value.

7) The fitting procedure needs to be rethought. It seems that the authors explore in a two-step approach a grid of values and parameter sets. First of all, where is the decision to run the simulation three times coming from? And how is the grid of parametric space explored? And also how are the values in the grid selected? What is the criteria? From Table 2 I notice some very weird combinations of parameters going from step 1 to step 2. It does not seem we are considering the whole parametric space in step 1 and then narrowing the ranges as a function of the results obtained in step 1. A much more comprehensive sensitivity analysis should be performed to evaluate more thoroughly the entire parametric space.

8) The best parameter sets are chosen based on a fit of the HZ incidence data, considered to be binomially distributed. Wouldn't it be better to consider a Poisson distribution for the HZ incidence data (13), considering that incidence refers to count of HZ episodes per age group?

9) The authors built an IBM that accounts for both between and within host dynamics, namely, taking into consideration both exogenous and endogenous boosting. The best parameter sets to inform the mathematical model are chosen based on repeated fitting of the HZ age-specific incidence estimates from the model to observed HZ incidence data from Belgium. As regards the fit of the best sets of parameters to the observed HZ data, as shown in Figure 1 and 2, the fit is very poor for the younger age groups, even for the two best parameter sets (9 and 13), as it is duly noted by the authors. Is there any way to improve the fit in this age group, considering that in the deterministic model by [13] it can be observed a better fit of HZ in young age groups? Maybe the authors should aggregate the incidence into age classes as otherwise the stochastic nature of the data might prevail over the goodness of fit.

10) It seems, from parameters sets 8, 9, and 13, that a higher peak fold increase following exogenous boosting is associated with a better fit of HZ incidence data in the first ten years of age. What about the age group between 10 and 20 years of age, the one suffering the poorest fit?

11) The authors claim that their model can easily accommodate new parameter value coming from experimental studies. However, when it comes to consider the effect of an hypothetic vaccination program, they choose a very simplistic and unrealistic one. Wouldn't it be better if they considered more realistic scenarios, such as those taken into consideration by [31] and other works, or some actual vaccination program, as the one in the US or in other countries with an effective varicella vaccination program? Also, the amount and timing of zoster reactivation may be country-specific as shown in [31]. This should be mentioned in the Discussion, especially considering the current debate on whether or not to introduce universal VZV vaccination in several countries.

12) Figures 4 and 5 should be redone considering maybe an aggregation over time, to avoid all the stochasticity of the model output. In Figure 4 the authors plot the pre-steady state and pre-vaccination incidence which is not of interest. And there is also a part of the plot without any line (right hand side). The interesting bit is the post vaccination part to which the authors only dedicate a small portion of the graph. Also, why are there different colors?

---

## [Author Response]

We believe that the individual comments by the reviewers do not point to major shortcomings of our methodology but mainly request clarifications of methods and results. Moreover, we believe some of the reviewers' comments are based on misconceptions, probably due to the paradigm shifting nature of our paper. We hope that *eLife* is willing to reconsider its position.

Reviewer #1:

*The main difficulty is that the model itself is hard to understand and therefore to assess what it adds to the existing literature. The Methods section plunges straight into a discussion of the estimation of the various parameters. It would be helpful to start with an overview of the model (with a written description and/or compartmental flow diagram), as well as the main ways it differs from previous models*.

We understand the demand for a better structuring and more elaboration. We have added a new figure detailing an important part of the model, we have majorly restructured the Methods section and we have added a new paragraph called “Model overview”.

*The comparison with previous models (which have been influential to vaccine policy in much of Europe) needs to be a bit sharper. In particular, it is not clear whether the magnitude of the HZ rise is smaller, the same or larger than previous models (e.g. Brisson). The fact that a perfect vaccine is used makes the comparison particularly difficult, although it is understandable why this was done in order to show effects more clearly. Hence even though the rise in HZ occurs earlier in life, it is not clear what the impact on cost-effectiveness will be*.

We would like to emphasize that our paper is not intended to be a new cost-effectiveness analysis, but instead to present a whole new framework allowing new cost-effectiveness studies to be performed. This is indeed the reason why we discussed the scenario of a perfect vaccine and why we have minimized the direct comparison of results with those resulting from using the other framework. However, we do clearly state that the total peak increase in HZ incidence is similar compared to the results from the other framework, but that a shift to younger age groups could have an important effect on morbidity.

*Also it should be noted that the conclusions are not completely independent of subjective model choices, since the form and sometimes the parameter values of functions representing cell-mediated immunity and boosting have to be pre-determined (albeit with reference to empirical work), as is inevitable. Hence it is not clear that this is the “final answer” to the exogenous boosting question*.

*However, the model clearly adds a great deal of additional insight to the existing debate, including the use of data that was not previously used in this fashion*.

We agree with the reviewer that subjective model choices could be the reason of bias. However, the fact that our model is based on experimentally observed data, in contrast to the ad hoc compartmentalization of the deterministic models, helps us in model selection.

Reviewer #2:

*[…] I have however some major points that should be carefully tackled. Authors criticize deterministic models used in current research in the field (Introduction, “One of the main points of criticism ... equilibrium states.”) Since however some of the main conclusions of this paper are in line with those of available deterministic models (e.g. in regards of the impact of vaccination) authors should explain clearly which are the possible gains allowed by IBMs, particularly in what respects the present approach should be superior to the deterministic models used in the literature. My main point here deals with identifiability of model parameters. The current model has at least 6 free parameters (i.e. more than in the currently used deterministic models), which as I understand are fitted to a single output, a zoster age-specific incidence profile. This should worse over-fitting problems, thereby emphasizing rather than reducing, the aforementioned criticism*.

We would like to note that the 2002 Brisson paper, that lies at the base of all Brisson-alike papers, clearly states in the Appendix that 11 parameters were estimated. However, we do acknowledge that Guzzetta and colleagues only use four parameters to estimate their reactivation rate. An important distinction however is the fact that they do not explicitly account for endogenous boosting for which we have reserved two parameters.

HZ incidence is not a scalar variable but a vectorized variable meaning that – in our case – 80 data points are taken into account for the goodness-of-fit (sufficient to estimate the seven parameters as also shown by our analyses). We do agree that future research should try and account for other fit possibilities and we suggest in our Discussion that new studies should try and obtain better VZV-CMI data as a function of age.

We have added the following text to the Discussion: “This means that in contrast to the deterministic models where abstract and ad hoc compartments are created to define the transition between different epidemiological states (for example the transition of a varicella recovered state to a zoster susceptible state), we can actually work with true biological, and verifiable, concepts such as VZV-CMI.” We also note that the following text was already present in the Discussion: “A major benefit of our modeling approach is the possibility to verify our best–fit parameter values via experimental studies, or vice versa, to adapt parameter values when empiricism delivers new insights. For example, if a new experimental study would prove the existence of endogenous boosting, this could be readily implemented in our VZV IBM so that parameter sets could be estimated, conditional on the existence of endogenous boosting. Although our IBM has given us valuable insights into between-host and within-host VZV dynamics, other influential factors could be introduced in future versions.”

*I am not entirely clear about the three hypotheses about CMI decrease after a boosting event (subsection “Modeling exogenous re-exposure to VZV”). In subsection “Modeling waning of VZV-CMI” a hypothesis has been made about CMI decline after primary CP infection. I believe one should take this – for reasons of both modeling parsimony and simplicity – as a baseline holding also for CMI decline after a boosting event, especially given that the data by Levin et al. used by the authors to parametrize the model did not seem to refer to CMI decline after primary chickenpox only. Other hypotheses could then naturally be grounded against this baseline*.

Levin et al. presented two types of data: (1) the decline of baseline (thus prior to vaccination) VZV-CMI as a function of age and (2) the decay of VZV-CMI after vaccination. The first is used as a starting value for the waning of VZV-CMI, whereas the second is used as a starting value for the decay of VZV-CMI after vaccination. Our model is thus based on these experimental findings. In order to avoid confusion we have added the notion in the Methods section that the waning of VZV-CMI is informed by the baseline evolution of VZV-CMI as a function of age.

*In particular I am not clear about the meaning of HP1, in subsection “Modeling exogenous re-exposure to VZV”, “Exponential decrease from peak to pre-boosting value based”*. *Why do you want to prevent that decline continues beyond the pre-boosting situation?*

We assume that boosting causes a temporary increase of VZV-CMI, so that when VZV-CMI has reached the “pre-boosting” value we can assume that the system has returned to its equilibrium state meaning that waning of VZV-CMI can continue. Figure 6 illustrates this concept. This is indeed an assumption and we hope that future VZV-CMI data will provide disclosure.

*The Hp3 formula is incomprehensible and needs careful explanation*.

We have changed delta_t to x and we have added “This function is constructed by the boundary conditions that Y(t = x) = Y_0_ and Y(t = 0) = P* Y_0_” to the explanation. This clarifies the equation (given the prior explanation that the formula represents an exponential decay).

*The force of boosting (subsection “Modeling exogenous re-exposure to VZV”) is assumed to occur at the same rate as the FOI. This assumption (which contrasts the authors' remark that the actual magnitude of the boosting effects is unknown) should be emphasized, as it might be a main responsible of the fact that seven of the best models “include exogenous boosting” (see Results)*.

The statement in the subsection “Modeling exogenous re-exposure to VZV” refers to a probability and not to a rate. We intentionally did not use the wording “force of infection” as this is a typical deterministic phrasing that is not applicable in this individual-based modeling setting. We followed the example of the progressive immunity model of Guzzetta et al. and assumed that all re-exposures to varicella resulted in effective boosting. Furthermore, the parameter values allow situations in which infectious contact occurs, without being boosted. These models, however, did not fit well to the data as none of these models were retained after our selection procedure.

*About the modeling of the Force of reactivation (subsection “Modeling VZV endogenous reactivation”), recent papers have modeled it as an increasing function of age (=senescence), and of time elapsed since last boosting episode. In the paper this is modeled as a purely random factor “The FoR deviates per time step and individual by means of a gamma function.” Am I understanding correctly? In my opinion that should be taken as a (useful) baseline against which, however, to ground alternative, possibly more realistic hypotheses. Therefore this point should be motivated more carefully. How was this gamma (density?) parametrized? This is important because one of the main results on the impact of vaccination i.e. the young age of zoster cases in the post-vaccination period (“Noteworthy is the relative dominance of the 31–40 year...”) might strongly follow from this hypothesis*.

First of all it is important to emphasize that the deterministic models use an epidemiological compartmentalization in which the force of reactivation represents the transition rate between two groups. In case of the deterministic models, this force of reactivation integrates the concepts of waning of immunity, VZV reactivation and endogenous boosting. And depending of the type of model, exogenous boosting might also be found in the equation for the reactivation rate (Brisson: No; Guzzetta: Yes). We use the force of reactivation as an explicit and independent characteristic of the behavior of VZV. It simply represents VZV reactivation. Whether it will evolve into zoster or endogenous boosting will depend on the value of VZV-CMI. This kind of modeling is not possible when using a deterministic framework. We have added “The reader should imagine the FoR to represent independent reactivation behavior of VZV and whether this reactivation will lead to HZ or endogenous boosting will depend on the value of VZV-CMI.” and “The FoR deviates per time step and individual by means of a gamma probability density function. This gamma function is chosen as it represents the summation of unknown biological phenomena that are assumed to have an exponential distribution.” to the paragraph concerning VZV reactivation. Furthermore, we have changed the title from “Modeling VZV endogenous reactivation” to “Modeling VZV reactivation”. In our VZV-IBM we have tested four different gamma distributions and these were included into the parameter set selection procedures. We reckon that our choice for a gamma distribution might be considered to be arbitrary, but we would like to emphasize that even in deterministic models arbitrary choices are made given the fact the exponential distribution is implicitly assumed for all these models (although often not explicitly mentioned).

Reviewer #3:

*[…] However, also considering the standard of the* eLife *journal, I believe the paper cannot be accepted in its current form. Some major improvements and changes are required to make it fit for publication. In particular, specific modeling assumptions and fitting procedures need to be carefully re-addressed and properly clarified before consideration for publication. See the list below for the required amendments*:

We believe we have made the necessary changes that allow our innovative paper to be suitable for publication in *eLife*.

*1) The analysis is based on an Agent-Based Modeling framework. These models require a very thorough validation procedure, especially when considering a realistic demographic population. The authors do not show any model output that convinces us that the demographic component is suitable to model VZV in Belgium. I believe that these validations outputs should be presented*.

We have completely reformatted our description of demography to “The dynamics of the synthetic population of the individual-based model are based on Belgian population and mortality data normalized to a fixed total population of 998,400 individuals [Eurostat, 2012]. The chosen population size is the result of a trade-off between ensuring sufficient heterogeneity and a manageable computational burden. Natural deaths are selected based on the age-dependent mortality rate. Per time step, the number of newborns equals the number of deaths to obtain a constant population size. To conduct predictions over many years, a stationary demographic structure is required throughout the simulated period. The Belgian population from 2012 has an overrepresentation of people from age 40 to 60 years (see Figure 7). Applying a fixed age-dependent mortality rate in combination with the replacement by newborns would results in an oscillating population distribution over time with respect to age. Therefore, the initial age distribution is not based on the Belgian population count from 2012 but on the survivor function estimated from data on the number of deaths per capita. The survivor function by age is estimated from data on the number of deaths per capita by m(a)=exp(−∫0aμ(t)dt) with age-dependent mortality rate *μ* using a thin plate regression spline model with the *gam-*function in the R-package *mgvc* (15, 39). Figure 7 illustrates the survivor function by age together with the Belgian population and mortality rates in 2012.

The model runs in time steps of one week and people with week-age 53 move to the next age class. To obtain homogeneous age transitions throughout the simulated period, initial week-ages are randomly assigned between 1 and 52. People from the last age class with 53 weeks are removed from the population so that the demographic structure remains stable throughout the simulated period.”

*2) Details of the model structure used are missing. Is the structure following the traditional Brisson/Edmunds/van Hoek approach or the Guzzetta progressive immunity one (AJE 2013)? Details on this should be available to the reader*.

We understand the demand for a better structuring and more elaboration. We have added a new figure detailing an important part of the model, we have majorly restructured the Methods section and we have added a new paragraph called “Model overview”.

3) The three exogenous boosting scenarios: a number of parameters are simply assumed and no explanations on their values are provided. In particular, why six weeks to reach the peak after exogenous boosting?

Our model is informed by experimentally observed data and most of the information is based on the 2008 Levin et al paper where the time point of six weeks was used as the time point to assess VZV-CMI. We agree that the time point of boosting could also be placed at four weeks after re-exposure, but on a larger scale this will have no effect on our results. We have added “The six week duration between the boosting event and the peak has been influenced by the Levin et al. data (18)” to the “Modeling exogenous re-exposure to VZV” paragraph.

*Also, in scenario 2, 60% of pre-boosting value (why?), steady state for x years (?) and why >= 3 years? A better explanation on these parameter values is required*.

Again, these numbers come from the Levin et al. paper. In order to further clarify this we have expanded our citation to that reference to “(cf. Figure 3 in the Levin et al. paper)”.

Also what is the biological explanation for the return to pre-boosting values after x years?

All vaccine studies show the tendency for cellular immunity after boosting to decay (in many studies after some time) to the base line values. Speculations in the experimental studies where this has been observed and in the biomedical literature in general, to try and explain this goes beyond VZV and certainly beyond the purpose of our paper. From a biological perspective this means that the original infection creates an equilibrium value (set-point) for VZV-CMI. From a clinical perspective this could also be illustrated by the decline of Zostavax vaccine efficacy with time after vaccination. However, we note that our parameter sets allow the “x” to be very large so that the duration of boosting could even go up to 20 years.

*Finally, in scenario 3 there's a little bit of confusion between delta t and x. The authors should check the formula of Y(t)*.

The reviewer is correct. We have changed all delta t’s to x.

*4) The lines “HZ individuals are assumed to be infectious for 1 week and receive a VZV-CMI reset to 1 +/− random factor (normally distributed, cf. primo infection) and “HZ only occurs in about 5% of individuals [*[40]*] and that VZV-CMI is higher in recovered HZ patients than in age-matched controls” seem contradicting. What is the post HZ value of VZV CMI? As after primary infection or higher?*

In our model successful VZV reactivation is assumed to cause an equal increase of VZV-CMI as compared to chickenpox. However, due to waning of VZV-CMI in individuals not having HZ the actual VZV-CMI could be higher in those after HZ.

*5) The sentence “Although HZ infectiousness was needed to maintain circulation of VZV in our model, the relative effects on overall VZV transmission are – compared to CP – marginal”: it is not clear here why you need HZ infectiousness to maintain circulation of VZV infection. If it is post-vaccination we don't care to maintain VZV circulation. If it is pre-vaccination, then there is something wrong if the model need HZ to keep VZV infection*.

Deterministic models do not need HZ infectiousness to maintain VZV transmission, but IBM do need HZ infectiousness (or multiple re-introductions from outside our populations, but that is beyond the scope of this paper) to reach endemic equilibrium (we realize that our model might be the only model so far to endemically model a viral infection by means of a IBM).

*6) The approximation of HZ infectiousness to varicella breakthrough is a strong assumption. at the least we need to check the sensitivity of model results to variations of this parameter value*.

We understand the hesitance of the reviewer regarding this approximation. We note that [7] used the same approximation. We added a sensitivity analysis to verify the robustness for our 13 best fitting parameter sets. This is shown as Figure 2–figure supplement 2. The reader can verify that for a range of parameter values of the HZ infectiousness (from 0.03 to 0.45) the predicted HZ incidence does not vary that much. We have also added “Figure 2–figure supplement 2 shows the results of a sensitivity analysis in which we varied HZ infectiousness from 0.03 to 0.45 for the thirteen best fitting parameter. As can be seen in Figure 2–figure supplement 2 our results are quite robust” to the respective Methods paragraph.

*7) The fitting procedure needs to be rethought. It seems that the authors explore in a two-step approach a grid of values and parameter sets. First of all, where is the decision to run the simulation three times coming from? And how is the grid of parametric space explored? And also how are the values in the grid selected? What is the criteria? From*
Table 2
*I notice some very weird combinations of parameters going from step 1 to step 2. It does not seem we are considering the whole parametric space in step 1 and then narrowing the ranges as a function of the results obtained in step 1. A much more comprehensive sensitivity analysis should be performed to evaluate more thoroughly the entire parametric space*.

Our results showed that the stochasticity of the simulations did not have a major influence on the predicted HZ incidences per parameter set as illustrated by the fact that the median relative variance of the deviance of the top 13 parameter sets for the three runs is only 1.9% (also see Figure 2–figure supplement 2). As such, we decided to use three runs per parameter set to account for a reasonable amount of stochasticity.

All parameter values are combined in the first parameter sets meaning that the entire parameter grid (informed by existing experimental data) is explored in the first step. In the second step we explore in more detail the most significant regions (as detailed in the Methods section). It is also important to realize that 1 simulation (thus one run for one parameter set) costs about 6-7 hours of computing time for one core meaning that it is not feasible to explore the entire parameter space with equal detail. Our presented results include data from more the 24,000 runs thus representing approximately 150,000 CPU hours (not including preparatory work).

*8) The best parameter sets are chosen based on a fit of the HZ incidence data, considered to be binomially distributed. Wouldn't it be better to consider a Poisson distribution for the HZ incidence data (*[13]*), considering that incidence refers to count of HZ episodes per age group?*

Published data by Yawn et al. showed that HZ recurrence rate within two years is 2% (even not significantly different from 1 when only looking at immunocompetent individuals). As such we assumed that individuals had either HZ or not during a year and this is best modeled by a binomial distribution.

*9) The authors built an IBM that accounts for both between and within host dynamics, namely, taking into consideration both exogenous and endogenous boosting. The best parameter sets to inform the mathematical model are chosen based on repeated fitting of the HZ age-specific incidence estimates from the model to observed HZ incidence data from Belgium. As regards the fit of the best sets of parameters to the observed HZ data, as shown in Figure 1 and 2, the fit is very poor for the younger age groups, even for the two best parameter sets (9 and 13), as it is duly noted by the authors. Is there any way to improve the fit in this age group, considering that in the deterministic model by*
[13]
*it can be observed a better fit of HZ in young age groups? Maybe the authors should aggregate the incidence into age classes as otherwise the stochastic nature of the data might prevail over the goodness of fit*.

We disagree with the strong statement that our model fits “very poor” for younger age groups. We also note that Figure 3 in Guzzetta et al. shows a poorer fit for the teenagers (although the size and scale of Figure 3 in Guzetta et al. hampers full appreciation of the fit). We do agree that our fit for the younger age groups is not as good as our excellent fits for the older – and more relevant – age groups and this finding might be due to a lack of relevant biological data for these age groups. We believe that future models and experimental studies should look into the effect of having chickenpox as an infant as it has been observed (as mentioned in our Discussion) that these individuals have lower VZV-CMI compared to others. We have expanded our sentence in the Discussion to “reduced VZV-CMI induction if infected during the first year of life as this might improve the prediction of the teenage group (34)”.

*10) It seems, from parameters sets 8, 9, and 13, that a higher peak fold increase following exogenous boosting is associated with a better fit of HZ incidence data in the first ten years of age*. *What about the age group between 10 and 20 years of age, the one suffering the poorest fit?*

Parameter sets 8, 9 and 13 also have other different parameters (including a higher VZV reactivation rate). As stated in the previous response, we believe that it is rather the observed HZ incidence in the teenage group that is higher than would be expected and we think that future versions of our model (in the same way that the Brisson models improved over the years) should try to account for this.

*11) The authors claim that their model can easily accommodate new parameter value coming from experimental studies. However, when it comes to consider the effect of an hypothetic vaccination program, they choose a very simplistic and unrealistic one. Wouldn't it be better if they considered more realistic scenarios, such as those taken into consideration by*
[31]
*and other works, or some actual vaccination program, as the one in the US or in other countries with an effective varicella vaccination program? Also, the amount and timing of zoster reactivation may be country-specific as shown in*
[31]*. This should be mentioned in the Discussion, especially considering the current debate on whether or not to introduce universal VZV vaccination in several countries*.

Our paper presents the first VZV IBM to date and presents a whole new method for dealing with the VZV “problem”, and estimates hitherto poorly documented immunological parameters. We agree with the comment by the reviewer that future more applied analyses should account for a broader (and more realistic) range of vaccination scenarios than the illustrative example from this conceptual paradigm shifting paper. We have added “If future VZV IBM are applied to project the impact of interventions, they should of course explore more realistic vaccination scenarios and assess inter-country variability in these explorations (31)” to the Discussion with reference to the Poletti et al. paper.

12) Figures 4 and 5 should be redone considering maybe an aggregation over time, to avoid all the stochasticity of the model output. In Figure 4 the authors plot the pre-steady state and pre-vaccination incidence which is not of interest. And there is also a part of the plot without any line (right hand side). The interesting bit is the post vaccination part to which the authors only dedicate a small portion of the graph. Also, why are there different colors?

Aggregation can be understood as presmoothing and it has been shown in the statistical literature that presmoothing is often not appropriate to handle stochasticity (see e.g. [Aerts 2010]). In order to present our results as transparent as possible we prefer to present the results “as they are”, meaning with the inclusion of full stochasticity. Otherwise the reader might be confused with the traditional output from deterministic models. We strongly disagree that the “interesting bit is post vaccination”. We have fitted this model to data pre-vaccination, and verify simultaneously some poorly understood biomedical parameters using a highly novel individual-based approach. The concept of the model is the primary interest in this paper, not the projections that can be made with it. We thus prefer to show all data including the pre-steady and pre-vaccination data. We only analyzed the post-vaccination data up to 80 years after vaccination.